# Emerging Intrinsic Therapeutic Targets for Metastatic Breast Cancer

**DOI:** 10.3390/biology12050697

**Published:** 2023-05-09

**Authors:** Jiawei Li, Eyleen L. K. Goh, Ji He, Yan Li, Zhimin Fan, Zhigang Yu, Peng Yuan, Dong-Xu Liu

**Affiliations:** 1The Centre for Biomedical and Chemical Sciences, School of Science, Faculty of Health and Environmental Sciences, Auckland University of Technology, Auckland 1010, New Zealand; 2Neuroscience and Mental Health Faculty, Lee Kong Chian School of Medicine, Nanyang Technological University, Singapore 308232, Singapore; 3Department of Breast Surgery, General Surgery Center, The First Hospital of Jilin University, Changchun 130021, China; 4Department of Breast Surgery, The Second Hospital of Shandong University, Jinan 250033, China; yuzhigang@sdu.edu.cn; 5Department of VIP Medical Services, National Cancer Centre/National Clinical Research Center for Cancer/Cancer Hospital, Chinese Academy of Medical Sciences and Peking Union Medical College, Beijing 100021, China

**Keywords:** breast cancer, targeted therapy, therapeutic target, metastatic breast cancer, immunotherapy, chemoimmunotherapy, histone deacetylase inhibitor, TROP-2, PARP, TNBC

## Abstract

**Simple Summary:**

Metastasis is the root cause of cancer death, responsible for roughly 90% of all cancer-related fatalities. Therefore, treating metastatic cancer is essential for optimal clinical management of these patients. Although there is currently no effective treatment for metastatic breast cancer, high-throughput approaches have identified novel molecules that are critical to tumor growth and metastasis in recent years. Novel therapies that target these molecules, such as immunotherapy, have been evaluated in both preclinical and clinical settings and have proven to be very promising in prolonging survival and relieving symptoms of metastatic disease, thereby enhancing the quality of life of patients. Due to the high intertumoral and intratumoral heterogeneity of cancer, the development of subtype-specific therapeutics and the use of combination treatments to simultaneously target multiple oncogenic signaling pathways may overcome drug resistance and achieve better clinical outcome for metastatic breast cancer.

**Abstract:**

Breast cancer is now the most common cancer worldwide, and it is also the main cause of cancer-related death in women. Survival rates for female breast cancer have significantly improved due to early diagnosis and better treatment. Nevertheless, for patients with advanced or metastatic breast cancer, the survival rate is still low, reflecting a need for the development of new therapies. Mechanistic insights into metastatic breast cancer have provided excellent opportunities for developing novel therapeutic strategies. Although high-throughput approaches have identified several therapeutic targets in metastatic disease, some subtypes such as triple-negative breast cancer do not yet have an apparent tumor-specific receptor or pathway to target. Therefore, exploring new druggable targets in metastatic disease is a high clinical priority. In this review, we summarize the emerging intrinsic therapeutic targets for metastatic breast cancer, including cyclin D-dependent kinases CDK4 and CDK6, the PI3K/AKT/mTOR pathway, the insulin/IGF1R pathway, the EGFR/HER family, the JAK/STAT pathway, poly(ADP-ribose) polymerases (PARP), TROP-2, Src kinases, histone modification enzymes, activated growth factor receptors, androgen receptors, breast cancer stem cells, matrix metalloproteinases, and immune checkpoint proteins. We also review the latest development in breast cancer immunotherapy. Drugs that target these molecules/pathways are either already FDA-approved or currently being tested in clinical trials.

## 1. Introduction

Breast cancer is the most common cancer in women, accounting for 24.5% of total cancer cases and 15.5% of cancer deaths [1]. Incidence rates of breast cancer in women are rising fast all over the world due to several risk factors, such as having fewer children, childbearing at late age, physical inactivity, and higher body mass index. However, the overall 5 year survival rate for breast cancer has now significantly improved to 91% from 63% in the early 1960s [2,3]. This trend may be the result of early diagnosis and better treatment. Nevertheless, for patients with advanced or metastatic breast cancer (abbreviated as mBC in this review) at the time of diagnosis, the 5 year survival rate stays at only 26%, reflecting a need for new insights and therapies into mBC [4].

Treatment for patients with breast cancer has been dramatically impacted by multiple newly approved drugs and indications. From 2010 to 2020, the United States Food and Drug Administration (FDA) approved 30 oncology drugs for patients with breast cancer. Of the 30 indications, 26 approvals were permitted in mBC, along with four in early breast cancer [5]. These drugs can be categorized into three classes: (1) cytotoxic chemotherapy, which kills proliferating cells primarily by interfering with mitosis [6]; (2) endocrine therapy or hormone therapy, which slows or stops the growth of female hormone-sensitive tumor cells; (3) targeted therapy, which kills cancer cells by inhibiting specific molecules controlling the growth and spread of tumor cells. Generally, targeted therapy is the basis of precision medicine. Most targeted therapies are either small-molecule drugs or monoclonal antibodies. Small-molecule drugs are agents small enough to easily enter cells and are more specific for cancer cells. Therefore, they are used as intracellular anticancer drugs [7]. Therapeutic monoclonal antibodies are designed to attach to specific antigens found on cancer cells. There are several aspects to be considered for therapeutic antibodies, including the selection of target antigens, immunogenicity of antibodies, penetration into solid tumors, half-life of antibodies, and ability of antibodies to recruit immune effector functions. The choice of target antigens play a crucial role in determining the success of antibody-based treatment [7].

With increasing understanding of the molecular biology of breast cancer, novel molecules that are critical to cancer cell growth and spread have been identified. There has been an elevated interest in the development of novel therapies that interfere with these molecules for higher precision of cancer treatment. In this review, we focus on the landscape of emerging tumor-intrinsic therapeutic targets for mBC.

## 2. Current Management of Breast Cancer

According to the expression of hormone receptors (HRs) and *ERBB2* gene amplification, breast cancer can be categorized into three subtypes. Approximately 75% of breast cancers that express estrogen receptor (ER) and/or progesterone receptor (PR) but not human epidermal growth factor receptor 2 (HER2) belong to the HR-positive (HR+)/HER2-negative (HER2−) breast cancer. Those that have *ERBB2* gene amplification with or without HR expression are called HER2-positive (HER2+) breast cancer, which accounts for 20–25% of all cases. The remaining 10–17% cases are triple-negative breast cancer (TNBC) that do not express any of the ER, PR, and HER2 proteins. In addition, the expression of the nuclear proliferation marker, Ki67, is widely used to measure cancer cell proliferation [8]. Recent progress in understanding the role of Ki67 in breast cancer has suggested its potential as a valuable prognostic biomarker [9]. Along with ER, PR, and HER2, the expression of Ki67 can assist oncologists in designing optimum treatment for patients [10]. 

These three subtypes have different risk profiles and treatment approaches. The best treatment for each patient is based on tumor subtype, anatomical cancer stage, and patient preference. For nonmetastatic breast cancer, the main purpose of treatment is to eradicate tumors from the breast and regional lymph nodes, and avoid metastatic recurrence, especially from reawakening of dormant/latent cancer cells [11]. For mBC, therapeutic purposes are to prolong life and symptom palliation [12]. This is because mBC currently remains incurable in almost all affected patients.

Treatment for nonmetastatic breast cancer mainly consists of local therapy and systemic therapy. Local therapy involves surgical resection and sampling, axillary lymph node removal, and postoperative radiation. The standard treatments are either a total mastectomy or an excision followed by radiation, which aims to achieve clear margins. These two surgical approaches have been shown equivalent in relapse-free and overall survival (OS) [13]. Systemic therapy can be postoperative (adjuvant) and preoperative (neoadjuvant). Subtype-directed breast cancer treatment comprises endocrine therapy for all HR+ tumors, HER2-targeted therapy plus chemotherapy for all HER2+ tumors, and chemotherapy for TNBC (Figure 1) [12].

Treatment for mBC can include endocrine therapy, chemotherapy, radiotherapy, biotherapy, bisphosphonates, or a combination of these therapies [14]. The treatments received are not for curative purposes. One of the main intents is to achieve the best possible quality of life for patients. Currently, systemic chemotherapy is the mainstay for mBC. However, chemotherapy alone reduces quality of life of patients because of its high toxicity. Chemotherapy combined with various targeted therapeutics has been shown to increase quality of life by reducing reliance on cytotoxic chemotherapy. As reviewed, combination therapies have demonstrated favorable results in terms of disease-free survival, progression-free survival (PFS), and overall response rates [15].

Clinically, patients with HR+ mBC are treated with endocrine therapy alone or in combination with cytotoxic chemotherapy. These patients normally have a favorable prognosis and are associated with better OS compared with other subtypes. However, a significant obstacle that limits the success of treatment is endocrine therapy resistance. Several factors, such as ERs and growth factors, contribute to endocrine therapy resistance [16]. ER gene (*ESR1*) mutations were recently identified as a rare cause of acquired endocrine resistance in HR+ mBC [17], indicating that second-line ER antagonists may be of substantial therapeutic benefit.

The more aggressive HER2+ mBC is characterized by high expression of HER2, which is encoded by the gene *ERBB2*. These patients are principally treated with monoclonal antibody trastuzumab (Herceptin), which became a landmark success in breast cancer targeted therapy and was the first FDA-approved drug for HER2+ breast cancer patients in 1998 [18]. Trastuzumab binds to domain IV of the extracellular domain of the HER2 receptor and inhibits homodimerization with other HER family receptors, thereby preventing HER2-mediated signaling in cancer cells. Trastuzumab kills HER2+ cells mainly by inducing antibody-dependent cellular cytotoxicity.

Pertuzumab is another monoclonal antibody against HER2. Recent studies found that the event-free survival of patients achieving a pathological complete response with pertuzumab plus trastuzumab was numerically better than with trastuzumab alone [19]. Pertuzumab can bind to a distinct extracellular epitope of HER2 (domain II) to prevent HER2 dimerization. Pertuzumab obtained FDA approval on 20 December 2017 [20]. In a phase III randomized clinical trial (APHINITY) of 4805 patients with stage I–III HER2+ breast cancer, pertuzumab statistically significantly improved the 3 year invasive disease-free survival (iDFS) [21]. The follow-up results confirmed that, in patients with early-stage breast cancer with positive HER2 lymph nodes, adding pertuzumab to standard trastuzumab/chemotherapy adjuvant therapy improved the iDFS (88% and 83% for pertuzumab and placebo, respectively) [22].

Neratinib, a small-molecule inhibitor of tyrosine kinase approved by the FDA on 25 February 2020 [23], can target several HER family members, including HER2. It was based on the results of a prospective randomized phase III trial (NALA), which found that neratinib in combination with capecitabine significantly improved the PFS, with fewer patients requiring intervention for central nervous system metastasis compared with lapatinib (a reversible dual tyrosine kinase inhibitor) plus capecitabine treatment for patients with HER2+ mBC [24]. Another randomized phase III ExteNET trial of 2840 HER2+/HR+ early-stage breast cancer patients compared 1 year of adjuvant daily neratinib with placebo after neoadjuvant and/or adjuvant therapy with chemotherapy and trastuzumab [25]. This trial demonstrated that 1 year of extended adjuvant therapy with neratinib improved the iDFS at 5 years. Compared with the pertuzumab adjuvant trial (APHINITY) that was consistent regardless of HR status [21], it is unclear why the iDFS advantage of this trial was noted only in the HR+ subgroup but not in the HR− group [25]. The final results of this trial further demonstrated that neratinib drastically improved the iDFS in patients with HER2+/HR+ tumors who received neratinib treatment within 1 year of prior trastuzumab-based therapy, and a similar trend was seen in patients with residual diseases after neoadjuvant therapy. The numerical improvement of central nervous system events and OS were consistent with the benefits of iDFS, thus indicating the long-term benefits of neratinib in this patient population [26]. However, many of these patients still relapsed and even died from breast cancers. Therefore, new approaches to targeting HER2 or other HER family members still need to be developed [27].

Although TNBC only accounts for a small percentage of all breast cancers, it is the most difficult type to treat. Treatments that target hormone receptors (ER and PR) and HER2 are not effective for TNBC because the drug targets do not exist [28]. General treatment for metastatic TNBC includes surgery, neoadjuvant or adjuvant cytotoxic chemotherapy, and radiation therapy. Innovative and more effective treatments for TNBC are always required. Novel therapies that target poly-adenosine diphosphate (ADP)-ribose polymerases (PARPs), human trophoblast cell-surface antigen 2 (TROP-2), and immune checkpoint proteins are already in use in clinics for TNBC. The mutations of breast cancer susceptibility gene 1 (*BRCA1*) and 2 (*BRCA2*) have been confirmed to serve as potential actionable targets for the treatment of TNBC patients [29]. Proteins encoded by *BRCA1* and *BRCA2* genes play a vital role in the repair process of DNA double-strand breaks. Mutations of *BRCA1* and *BRCA2* genes significantly impact the DNA repair system, accounting for approximately 2–3% of breast cancer events and over 10% in TNBC [30]. The clinical efficacy of targeted therapy associated with these DNA alterations has been confirmed in TNBC patients [29]. As discussed below, there are no more potential treatments being developed for TNBC.

## 3. Tumor Intrinsic Targeting

Tumor-intrinsic signaling pathways play a pivotal role in tumorigenesis. They drive uncontrolled cell-cycle progression and trigger apoptosis or senescence. These contingent processes rely on a series of sensors and transducers. Oncogenic mutations in these pathways may have far-reaching and different effects on the evolution of tumors. Therefore, they may leave a complete potential for tumor suppression and become therapeutic targets for cancer patients [31]. In this section, we discuss promising therapeutic tumor intrinsic signaling candidates for breast cancer treatment (Table 1). We also summarize the latest development in immune therapy for TNBC.

### 3.1. Targeting CDK4 and CDK6 in Breast Cancer

Maintaining tissue homeostasis depends on two key physiologic processes: cell division and death, which are tightly controlled in the cell cycle. The cell-cycle process consists of four ordered phases, which are G0/G1 (gap 1), S (DNA synthesis), G2 (gap 2), and M (mitosis) [32]. CDK4 and CDK6, two serine/threonine kinases, can regulate the mid-G1 phase. The catalytic activity of CDK4/6 is regulated by cyclin D (D1, D2, and D3) [33]. At the start of G1, retinoblastoma (RB)-associated proteins are active but non-phosphorylated [34]. RB can bind to the transcription factors of E2F family and repress the transcription of E2F-regulated genes, resulting in the inhibition of cell-cycle progression, DNA replication, and mitosis [35]. After cyclin D binds to CDK4/6, the complex of cyclin D and CDK4/6 selectively phosphorylates RB protein (pRB) and inactivates pRB’s inhibitory ability [36,37]. The kinase activity of CDK4/6 is counterbalanced by the INK4 family of CDK4/6 inhibitors, which inhibit the kinase activity of CDK4/6 by preventing cyclin D from binding to CDK4/6 [34].

*RB* is a tumor suppressor. The cyclin D/CDK/pRB pathway plays a critical role in the development of mBC. Upon mitogenic and oncogenic stimulation, the expression of D-type cyclins increases, resulting in activation of CDK4/6 kinases and subsequent phosphorylation of RB proteins. Phosphorylated RB is inactive, releasing the E2F family of transcription factors, which relieves its transcriptional repression and allows for cell-cycle progression of cancer cells [38]. Deregulation of the cyclin D/CDK/pRB pathway is frequently observed in breast cancer in two main ways. First, pRB loss is much more frequent in TNBC than in the other breast cancer subtypes [39]. Second, the expression of pRB in TNBC is significantly associated with sporadic TNBC and metastasis to bone [40].

CDK4/6 are positive regulators of cell-cycle entry, and they are overactive in most human cancers. Targeting CDK4/6 with specific inhibitors in HR+ mBC has been confirmed effective in both clinical and preclinical settings. Recently, the FDA and the European Medicine Agency (EMA) approved three highly selective oral inhibitors of CDK4/6 for HR+ mBC: palbociclib, ribociclib, and abemaciclib [32]. The phase III PALOMA 3 trial, which recruited 521 HR+ HER2− mBC patients, demonstrated that patients treated with the combination of palbociclib and fulvestrant benefited with longer OS than patients treated with fulvestrant alone [41]. The trials of ribociclib also showed that, in patients with HR+ HER2− mBC, the OS rate of patients receiving CDK4/6 inhibitors plus endocrine therapy was significantly higher than that of patients receiving endocrine therapy alone. No new concerns on toxic effects emerged over a longer follow-up period [42,43]. Similarly, the phase III MONARCH 2 trial found that abemaciclib plus fulvestrant substantially improved the median OS for patients with ER+ HER− mBC [44]. Despite the success in both ER+ and HER+ disease, the efficacy of these CDK4/6 inhibitors is still limited due to drug resistance. Research should next focus on delineating the cellular mechanism of CDK4/6 promoting tumorigenesis and identifying specific subtypes, such as pRB-positive TNBC [40], which are sensitive to the treatments to improve clinical outcomes.

### 3.2. Targeting the PI3K/AKT/mTOR Pathway in Breast Cancer

The phosphoinositide 3 kinase (PI3K)/AKT/mammalian target of rapamycin (mTOR) pathway plays a critical role in cell growth, tumor proliferation, and therapy resistance in breast cancer [45]. The PI3K heterodimer belongs to the class IA family of PI3Ks and plays a key role in this pathway. It consists of two subunits, a regulatory subunit (p85) and a catalytic subunit (p110) (Figure 2). It has been shown that p85 can regulate p110 activation in the absence or presence of upstream stimulation by growth factor receptor tyrosine kinases (RTKs) [46]. Each subunit has a different set of encoding genes; p85 is encoded by *PIK3R1*, *PIK3R2*, or *PIK3R3*, while p110 by *PIK3CA*, *PIK3CB*, or *PIK3CD* [47].

Acquisition of *PIK3CA* mutations, which usually exist in exons 9 and 20, is the most common abnormality in human malignant tumors, including breast cancer [48]. Several groups initiated a large number of analyses and demonstrated that *PIK3CA* mutations were most commonly found in breast cancer [49]. *PIK3CA* mutations have also been found to be associated with the response to breast cancer treatment. It has been demonstrated that oncogenic *PIK3CA* mutations may contribute to more resistance to antibody-based therapeutic trastuzumab and anti-HER2 agent lapatinib (a dual tyrosine kinase inhibitor blocking HER1 and HER2 tyrosine kinase activity) treatment in breast cancer [50,51]. In contrast, there is no association between *PIK3CA* mutational status and sensitivity to standard chemotherapy [52]. In 2019, the FDA approved a p110α-specific PI3K inhibitor alpelisib (Piqray) for breast cancer treatment in combination with endocrine therapy. Treatment using alpelisib together with fulvestrant showed excellent efficacy in postmenopausal women with *PIK3CA*-mutated, HR+/HER2− mBC patients who had previously received endocrine therapy [53]. Regulatory authorities in Europe and Australia have now approved alpelisib to be used with endocrine therapy for patients with HR+/HER2−, *PIK3CA*-mutated mBC [54].

Another three PI3K inhibitors for treating *PIK3CA*-mutated mBC, namely, buparlisib, pictilisib, and taselisib, are still under development in clinical trials. Buparlisib is an oral pan-PI3K inhibitor. Its safety and efficacy in combination with fulvestrant have been evaluated in patients with HR+/HER2− mBC in two phase III randomized clinical trials, BELLE-2 and BELLE-3 [55,56]. Unfortunately, some patients had poor tolerance to buparlisib and were suspended from the treatment [55], but both trials achieved their primary efficacy endpoint. Moreover, a phase II FERGI clinical trial for patients with HR+/HER2− mBC was conducted to evaluate the safety and efficacy of pictilisib when administered with fulvestrant. The results showed that the tolerability and efficacy of pictilisib were limited by its toxicity [57]. Taselisib can equipotently inhibit p110 and is more sensitive to p110α mutation compared with the wildtype isoform. The phase III SANDPIPER clinical trial evaluated the treatment of taselisib plus fulvestrant for mBC patients with recurrence or progression after primary treatment. This trial met its primary endpoint with improved PFS in patients with *PIK3CA*-mutant tumors (7.4 months in the taselisib versus 5.4 months in placebo arm). However, the combination of taselisib plus fulvestrant offered limited benefits and tolerability [58]. Therefore, more safety and efficacy studies on these inhibitors are needed for further clinical uses.

Since its discovery, the mTOR molecular pathway has been implicated in the pathogenesis of breast cancer and become a prominent therapeutic target in addition to targeting *PIK3CA* mutation [59]. mTOR is a vital signaling integrator related to cell-cycle progression. Inhibition of mTOR mainly leads to activation of downstream protein kinases required for ribosomal biosynthesis and interruption of mRNAs translation, resulting in the failure of cell-cycle transition from G1 to S phase [60,61]. Everolimus, an mTOR inhibitor, can bind to its intracellular receptor FKBP12 with high affinity, which belongs to the family of immunophilin proteins [62]. The complex of everolimus and FKBP12 inhibits mTOR activity, preventing the binding of raptor to mTORC1 and its downstream signaling, thus resulting in the inhibition of cell growth and proliferation [63]. A clinical trial evaluated everolimus ± endocrine therapy in selected HR+ mBC patients, demonstrating that the combination of everolimus and endocrine therapy significantly improved PFS when compared with endocrine therapy alone, with a median PFS of 10.6 months and 4.1 months, respectively [64,65]. In 2012, the FDA approved the treatment of everolimus combined with exemestane for mBC patients who have acquired resistance to hormone therapies [59]. Dactolisib, codenamed NVP-BEZ235 and BEZ235, can inhibit both PI3K and mTOR. It is being considered as a potential therapeutic target for treating breast cancer. Preclinical studies in transgenic xenograft mouse tumor models claimed that dactolisib inhibited the growth of breast cancer tumors effectively [66]. However, a phase IB clinical trial enrolled with 19 patients with advanced or metastatic solid cancers only exhibited limited efficacy and tolerance when patients were treated with the combination of dactolisib and everolimus, possibly due to drug–drug interactions [67]. Nonetheless, it is worth noting that, in two independent orthotopic xenograft rat and mouse models of glioblastoma, dactolisib treatment showed no survival benefit or inhibition of tumor growth, while severe side-effects, such as elevated levels of blood glucose and the liver enzyme alanine transaminase, were observed [68]. The company Novartis, who owned dactolisib, stated in its annual report filed in January 2014 that it had discontinued development of dactolisib in oncology indications.

### 3.3. Inhibiting PARP in TNBC

The poly-adenosine diphosphate (ADP)-ribose polymerase (PARP) multifunctional enzymes play a vital role in the repair system of DNA breaks caused by *BRCA* gene mutations [30]. PARP-1 is more involved than PARP-2 and PARP-3 in the DNA repair process. PARP-1, activated by DNA damage, can bind to the nearest position of DNA break and trigger the base excision repair system [69]. Therefore, the inhibition of PARP enzymes contributes to the accumulation of DNA breaks. It has been shown that PARP inhibitors can specifically target *BRCA*-mutated cells [70]. Olaparib (Lynparza) is one of the most studied PARP inhibitors. In a phase III clinical trial for HER2− mBC patients with *BRCA* mutation, olaparib was reported with positive results [69]. Another recent study of 302 patients with HR− mBC and *BRCA* mutation, divided into two groups (205 patients receiving olaparib and 97 in standard therapy), showed that median PFS was significantly longer with oral olaparib monotherapy than with standard chemotherapy [70]. Since the introduction of cytotoxic chemotherapy, olaparib is the first new medication with excellent benefits in metastatic TNBC [71]. Over 30 actively recruiting breast cancer trials involve olaparib. In January 2018, the NCCN guidelines added olaparib to the guidelines of breast cancer treatment and Category 1 recommendations [71]. The efficacy of olaparib was fully investigated in a large-scale phase III, double-blinded, randomized trial OlympiA with 1836 patients who had HER2-negative/*BRCA*-mutated tumors and had received local treatment and neoadjuvant or adjuvant chemotherapy [72]. The interim analysis of this trial clearly found that adjuvant olaparib was associated with significantly longer survival: the 3 year invasive disease-free survival was 85.9% in the olaparib group vs. 77.1% in the placebo group, and the 3 year distant disease-free survival was 87.5% vs. 80.4%, respectively [72]. On the basis of this result, on 11 March 2022, FDA approved olaparib for the treatment of *BRCA*-mutated, HER2-negative high-risk early breast cancer patients who have received primary treatments [73].

Other PARP inhibitors, such as talazoparib (Talzenna), veliparib, niraparib (Zejula), and rucaparib (Rubraca), are either under or have completed clinical trials to verify their role in treating breast cancer patients. The EMBRACA trial with 431 *BRCA*-mutated mBC patients evaluated the safety and efficacy of talazoparib compared with standard chemotherapy [74]. In this trial, patients receiving talazoparib (*n* = 287) achieved longer median PFS than patients with chemotherapy (*n* = 144); outcomes reported by patients were superior with talazoparib [74]. On the basis of these interim results, talazoparib was approved by the FDA for treating *BRCA*-mutated HER2− mBC patients on 16 October 2018 [75]. However, the final overall survival analysis of the EMBRACA trial did not identify any benefit of talazoparib treatment over chemotherapy for these patients [76].

Another clinical trial of PARP inhibitors was the BrighTNess phase III trial, which evaluated the efficacy of the addition of the PARP inhibitor veliparib plus carboplatin or carboplatin alone to standard neoadjuvant chemotherapy in TNBC [77]. From 4 April 2014 to 18 March 2016, 634 patients from 146 locations in 15 countries were recruited and randomly assigned. Primary findings showed that patients treated with paclitaxel, carboplatin, and veliparib achieved a higher pathological complete response than patients who received paclitaxel alone, but there was no difference compared with patients who were treated with paclitaxel plus carboplatin [77]. These results suggested that the addition of both veliparib and carboplatin to paclitaxel had a benefit, but the addition of veliparib alone to carboplatin and paclitaxel did not. A longer (4.5 year) follow-up of the BrighTNess trial also failed to observe any impact of adding the PARP inhibitor veliparib to carboplatin-containing neoadjuvant chemotherapy on the long-term outcomes in TNBC patients [78].

Because *BRCA*-mutated breast cancers are deficient in homologous recombination repair of DNA damage, they are sensitive to PARP inhibitors and platinum agents. A randomized, double-blinded, placebo-controlled, phase III trial (BROCADE3) of 2202 patients was conducted at 147 hospitals in 36 countries between 30 July 2014 and 17 January 2018, to assess the efficacy of the PARP inhibitor veliparib in the 513 eligible patients with *BRCA*-mutated HER2-negative mBC [79]. This trial demonstrated that addition of veliparib to a highly active platinum doublet (carboplatin–paclitaxel), with continuation as monotherapy if the doublet was discontinued, significantly improved the median PFS from 12.6 months in the carboplatin-paclitaxel control group to 14.5 months in the veliparib plus platinum doublet group [79]. This trial supported the utility of combining platinum and PARP inhibitors in patients with *BRCA*-mutated HER2-negative mBC. Moreover, a number of subgroup analyses of the BROCADE3 trial confirmed the benefit of combining veliparib with carboplatin–paclitaxel for this patient population irrespective of the status of HR or germline *BRCA1/2* mutations [80], whether veliparib was used as first-line therapy [81] or maintenance monotherapy [82]. However, it was observed that crossover veliparib monotherapy, i.e., patients were treated with veliparib monotherapy after disease progression on placebo plus carboplatin/paclitaxel, had limited antitumor activity in the 513 patients in the BROCADE3 trial [83].

The safety of PARP inhibitors might be an issue. Despite the exposure safety analysis not identifying any meaningful exposure-dependent trend in the incidence of adverse events of interest that was associated with the dose regimen of veliparib used in the BROCADE3 trial [84], three PARP inhibitors—niraparib, olaparib and rucaparib—were voluntarily withdrawn by the manufacturers for heavily pretreated, *BRCA*-mutated advanced ovarian cancer due to safety concerns. For instance, on 26 August 2022, the manufacturer AstraZeneca of olaparib released the results of an OS subgroup analysis of the phase III SOLO3 trial, stating that “a potential survival detrimental effect” on the OS was observed in the ovarian cancer patients who received olaparib after at least two lines of platinum-based chemotherapy compared to the chemotherapy control. As a result, olaparib was withdrawn from the indication.

### 3.4. Targeting TROP-2 in Breast Cancer

TROP-2 is a transmembrane glycoprotein and functions as an intracellular calcium signal transducer. TROP-2 signaling is involved in cancer growth, invasion, and metastasis via interaction with several ligands, including claudin-1, claudin-7, cyclin D1, and potentially IGF-1 [85]. TROP-2 is overexpressed in almost all epithelial cancers compared with normal tissues [86,87]. A recent gene expression analysis found that TROP-2 is expressed in all breast cancer subtypes, particularly luminal A and TNBC, and correlated with the expression of many genes that are involved in cell epithelial transformation, adhesion, and proliferation [88]. Moreover, the upregulation of TROP-2 in cancer cells has been implicated in drug resistance to the antiestrogen drug tamoxifen and the anti-HER2 monoclonal drug trastuzumab [89].

TROP-2 is now a rising star in the development of anticancer agents [90]. Metastatic cancer tissues have an elevated level of TROP-2 expression, suggesting that it is a potential therapeutic target for late-stage diseases [91]. A recent study evaluated an antibody–drug conjugate (ADC), sacituzumab govitecan, which targets TROP-2 for selective transmission of the active metabolite SN-38 of the chemodrug irinotecan to cancer cells in metastatic TNBC patients [92]. The study found that sacituzumab govitecan is well tolerated and induces early and lasting responses in metastatic TNBC patients who have received a large number of pretreatments [92]. Moreover, the phase III randomized ASCENT study demonstrated that PFS in patients with metastatic TNBC is significantly improved with sacituzumab govitecan compared with single-agent chemotherapy [93]. On the basis of the results of the ASCENT study, this drug was approved by the FDA on 7 April 2021 for metastatic TNBC that received two or more prior primary therapies [94].

In addition to TNBC, the therapeutic potential of sacituzumab govitecan has been assessed in other types of breast cancer [95]. Studies have demonstrated that sacituzumab govitecan treatment can also improve outcomes of patients with metastatic HR+/HER2− breast tumors. In the multicenter, open-label, randomized, phase III trial (TROPiCS-02), the efficacy of sacituzumab govitecan in the treatment of HR+/HER2− mBC was investigated [96]. It was found that sacituzumab govitecan significantly improved both PFS and OS as compared with single-agent chemotherapy [96]. On the basis of these results, on 3 February 2023, the FDA approved sacituzumab govitecan (-hziy) for patients with HR+, HER− mBC who received prior endocrine-based therapy and at least two additional systemic therapies in the metastatic setting [97].

A number of other anti-TROP-2 ADC agents are currently being investigated in preclinical or clinical trials, with datopotamab deruxtecan (Dato-DXd; DS-1062) and SKB264 being the front runners [98]. Dato-DXd is different from sacituzumab govitecan in that it has a cleavable tetrapeptide linker and a more potent topoisomerase inhibitor payload [95]. A phase I study (TROPION-PanTumor01) evaluated the effect of Dato-DXd on unselected, previously treated patients with advanced TNBC and HR+/HER2− mBC [99]. This study showed promising results that prompted two phase III clinical trials: the TROPION Breast01 [100] and the TROPION-Breast02 [101] trials. The former is a study of Dato-DXd vs. chemotherapy in HR+, HER2− mBC patients, while the latter evaluates TNBC patients. SKB264 is another ADC composed of an anti-TROP-2 antibody coupled to the cytotoxic belotecan derivative via a novel linker. A phase III study of SKB264 in metastatic TNBC was also initiated [102]. Of note, two TROP-2-directed ADCs, namely, PF-06664178 (RN927C) from Pfizer and BAT8003 from Bio-Thera failed to achieve a sufficient therapeutic window in phase I clinical trials [103].

There are studies that evaluated the benefit of TROP-2 inhibition in combination with immunotherapy, targeted therapy, and other currently available treatments [104]. We envisage that there will be other strategies, such as bispecific antibodies, virus-like particles, and nanoparticles, to be developed as a new class of novel therapeutics to target TROP-2 in the near future [105].

### 3.5. Targeting Src Kinases in Breast Cancer

The oncogene *Src* plays a key role in many important cellular pathways. The Src family kinases (SFKs) are regulatory proteins, involved in cell survival, proliferation, differentiation, invasion, and motility [106]. Increased expression of Src has been observed in many solid tumors, including breast cancer [107]. Interruption of Src signaling can disrupt various oncogenic pathways, contributing to slower disease progression and preventing cancer recurrence and metastasis [108]. Src has been identified as a potential therapeutic target for breast cancer. Several Src inhibitors have been developed by drug companies [109]. Currently, a combination of Src inhibitor and chemotherapy, with/without trastuzumab, has been studied in breast cancer patients. However, a few phase II studies of Src inhibitor monotherapy failed to achieve a considerable antitumor activity [110,111].

Saracatinib, an oral Src inhibitor of Src-associated signaling [112], can bind to the activated ATP binding site to inactivate Src signaling. Preclinical data in an HR− breast cancer cell line and xenograft models demonstrated that saracatinib significantly decreased cell growth and migration [113]. Saracatinib was also reported to inhibit Src phosphorylation and cell growth in the MCF-7 (HR+) human breast cancer cell line [114]. However, a phase II clinical trial, which evaluated the efficacy of saracatinib in HR− mBC patients, did not support saracatinib as a monotherapy for the treatment of ER−/PR− mBC because no patients achieved any complete or partial response [115].

Dasatinib is another promising inhibitor of Src signaling. It was initially approved by the FDA for the treatment of special subset of leukemia, but not for breast cancer until recently [116]. Preclinical studies have demonstrated that dasatinib can inhibit breast cancer cell growth [117], but the clinical efficacy of this drug still lacks evidence. A phase II trial, which evaluated the efficacy and safety of dasatinib alone in patients with advanced TNBC, indicated limited activity in unselected patients with TNBC [111]. Another phase II study also obtained a similar result that dasatinib alone failed to exhibit significant antitumor activity in patients with heavily pretreated mBC [110]. Nevertheless, a phase II study involving 40 HR− mBC patients reported promising benefits from the combination of dasatinib with paclitaxel [118]. Although the trial stopped early due to slow accrual, it was suggested that combined therapies may work better in these patients [118].

Another Src inhibitor, bosutinib, was demonstrated to inhibit breast cancer cell proliferation, invasion, and migration in preclinical studies [119]. The efficacy and safety of bosutinib were further assessed in pretreated patients with local mBC [120]. The study indicated that bosutinib showed promising efficacy in prolonging time to progression in chemotherapy pretreated patients with local mBC; it was generally well tolerated, but had a different safety profile from dastinib in a similar patient population [120]. Dasatinib elicited much higher rates of hematological complications (such as anemia and neutropenia) than bosutinib [120].

### 3.6. Targeting the IL-6/JAK/STAT Pathway in Breast Cancer

The Janus kinase/signal transducers and activators of transcription (JAK/STAT) signaling pathway regulates cellular responses to growth factors and cytokines and is regulated through many protein interactions [121]. Activation of JAK and STAT signaling is often observed in breast cancer [122]. Phosphorylation of STAT proteins through activated JAKs causes nuclear translocation and transcriptional regulation, resulting in cell proliferation, apoptosis, and pathogenesis [123]. Mutations in JAK and STAT proteins are linked to constitutive activation of JAK/STAT signaling [124]. Therefore, it is possible to target JAK/STAT signaling for the treatment of cancer.

The STAT family comprises seven structurally similar and highly conserved members: STAT1-4, STAT5a/b, and STAT6. STAT3 was initially identified to bind to DNA in response to cytokine interleukin-6 (IL-6) and epidermal growth factor (EGF) in 1994 [125,126]. STAT3 can also be activated by IL-17 [127]. However, STAT3 was reported to be downregulated by IL-32θ, an IL-32 isoform, via direct interaction with PKCδ, resulting in a slowdown of the progression of macrophage-associated breast cancer [128]. It was also shown that the WW domain-containing oxidoreductase inhibits tumor growth and metastasis in TNBC by inhibiting JAK2 phosphorylation and impeding the association of JAK2 with STAT3, thereby preventing STAT3 phosphorylation [129]. Since the discovery of STAT3, the role of IL-6/JAK/STAT3 signaling has been widely studied in various cancers [130,131,132]. This signaling axis plays a vital role in tumor progression and metastasis [131,133]. Analyses of human breast tumor samples demonstrated that the expression of IL-6 is elevated in the early stage of breast tumors and is positively associated with advanced tumor stages, suggesting a critical role in breast tumor metastasis [134].

The Drosophila ortholog of GRAMD1B was initially identified as a signaling component of the Drosophila JAK/STAT pathway [135]. The expression of GRAMD1B is significantly increased upon IL-6 stimulation. In contrast, JAK2 inhibitor AG490 can downregulate the expression of GRAMD1B in breast cancer MDA-MB-231 cells, suggesting that JAK/STAT signaling controls the expression of GRAMD1B in breast cancer cells [136]. GRAMD1B was shown to inhibit breast cancer cell proliferation and promote cell death by deactivating JAK/STAT signaling [136]. However, the role of GRAMD1B in breast cancer is still largely unknown.

Monoclonal antibodies and small inhibitory molecules targeting the IL-6/JAK/STAT3 pathway, including agents that interfere with IL-6 and its receptor or JAKs, have already received FDA approval for the treatment of inflammatory conditions or myeloproliferative neoplasms [137]. Although the FDA has not approved any IL-6/JAK/STAT3 pathway inhibitors for breast cancer, many of them are currently under development in preclinical and clinical investigation [132].

### 3.7. Histone Deacetylase Inhibitors for Breast Cancer

Histone deacetylase inhibitors (HDACis) are a relatively new type of drug that can modulate cell death, apoptosis, and cell-cycle arrest in cancer [138]. HDACis can enhance the acetylation of cellular proteins by blocking the activity of histone deacetylases (HDACs) [139], thus inhibiting the growth of tumors and apoptosis of cancer cells, whereas normal tissue is not particularly impacted [140]. The rationale for targeting HDACs in cancer therapy is as follows: altered HDAC expression and function are frequently observed in many cancers. Disrupted acetylation homeostasis in cells is considered to contribute to tumorigenesis. Because acetylation is reversible via HDACs, they can be exploited for cancer treatment. HDACs reversibly modify the acetylation status of histones and non-histones, which results in a genome-wide change in gene expression without affecting the DNA sequences. On the other hand, HDACis counteract the abnormal acetylation found in cancer cells, restore the expression of tumor suppressors, induce cell-cycle arrest, apoptosis, and differentiation, and inhibit angiogenesis and metastasis [140]. Moreover, it was also observed that cancer cells are more sensitive to HDACi-induced apoptosis than normal cells [141], thus strengthening the therapeutic potential of HDACis.

Preclinical studies have demonstrated that HDACs modulate ER activity, and HDACis reverse resistance to antiestrogen therapies for breast cancer in vitro [142]. The most commonly used HDACis in clinical research are entinostat, vorinostat, and panobinostat. Entinostat was demonstrated to increase ER expression and induce the re-expression of androgen receptor (AR) and aromatase enzymes, which may benefit patients with ER-negative and endocrine-resistant breast cancers [143]. It was also observed that the combination of entinostat and the HER2-targeting drug (lapatinib) enhanced the antitumor effect of lapatinib in HER2+ breast cancer cells [144]. Vorinostat has been shown to inhibit cell proliferation and induce cell death in TNBC cell lines in combination with the PARP inhibitor olarparib [145]. Vorinostat was also shown to have antiproliferation effects and regulate tumor growth in a TNBC cell line-derived xenograft in mice [146]. Furthermore, the pan-HDACi panobinostat was shown to increase the expression of E-cadherin and decrease cell invasion and migration without affecting the estrogen pathway in TNBC cells, suggesting a possible use of panobinostat for mBC patients who are refractory to hormonal therapy [147]. Another mechanism for this pan-HDACi was also revealed in breast cancer mouse models. Panobinostat was shown to inhibit tumor growth and metastasis by downregulating the Wnt/β-catenin signaling pathway through upregulating the expression of the adenomatous polyposis coli protein-like (APCL) tumor suppressor gene in breast cancer cells [148].

So far, the FDA has approved several HDACis for treating various cancers, but no HDACis have been specifically earmarked for breast cancer treatment in clinics [149]. A large number of clinical trials of HDACis for breast cancer treatment have been conducted, and their results indicate that HDACis have antitumor function and may be clinically effective [139]. Three phase III studies have been designed to evaluate the efficacy of HDACis (entinostat and chidamide) combined with endocrine therapy for mBC, and 16 phase II studies have been conducted to explore the effects of HDACis or combined with chemotherapy on breast cancer. The remaining studies are at the phase I stage [139]. Although these HDACis seem to be promising for breast cancer management, the sample sizes of the completed studies were relatively small. Hence, more evidence is needed to warrant large-scale, multicenter clinical trials.

### 3.8. Inhibition of EGFR/HER2

Activated growth factor receptors (GFRs) play a vital role in cell proliferation, invasion, and metastasis in breast cancer. Overexpression of growth factor receptors has been observed to be correlated with poor clinical outcomes in breast cancer patients [150]. Therefore, new therapies can be developed to target these receptors to inhibit tumor growth and improve patient survival. The epidermal growth factor receptor (EGFR) is one of the first identified therapeutic targets in this category of targeted therapy. Overexpression of EGFR was observed in almost half of the cases of TNBC and inflammatory breast cancer, which are especially aggressive [151]. Approximately 15% to 20% of invasive breast cancers have amplification of *ERBB2* gene or overexpression of the HER2 protein [152]. Clinically, several drugs targeting HER2 receptors have been used for patients with HER2-overexpressing breast cancer. Trastuzumab was the first approved drug for HER2+ mBC patients. Novel trastuzumab therapy in combination with other therapies is being studied in mBC patients [150]. Lapatinib is also registered for the treatment of HER2+ mBC patients in combination with capecitabine, as well as for HR+ mBC patients in combination with an aromatase inhibitor [153]. These combined treatments were approved by the FDA on 13 March 2007 [154]. Another HER2 dual tyrosine kinase inhibitor, pyrotinib, demonstrated promising antitumor efficacy with tolerable toxicity in HER2+ mBC. Clinical trials have been conducted to evaluate the efficacy of pyrotinib for the treatment of breast cancer. In a randomized phase II trial from 29 May 2015 to 15 March 2016, 128 eligible patients with HER2+ mBC were included and randomly assigned to compare the efficacy of pyrotinib and lapatinib treatment. The results showed that pyrotinib and capecitabine statistically improved the overall response and PFS as compared with lapatinib and capecitabine [155]. Another clinical trial with 279 HER2+ mBC patients recruited between July 2016 and November 2017 also demonstrated that pretreated mBC patients achieved significantly better PFS by treatment with pyrotinib and capecitabine [156]. The more recent clinical trial from July 2017 to October 2018, which included 267 patients with HER2+ mBC, further confirmed that, in patients after trastuzumab and chemotherapy, pyrotinib plus capecitabine achieved significant better PFS than lapatinib plus capecitabine, with manageable toxicity, thus suggesting an alternative treatment option for HER+ mBC [157,158].

### 3.9. Targeting Insulin/IGF1R Signaling in mBC

The insulin/IGF1R signaling pathway is observed to be activated in 75% of breast cancers and 87% of invasive breast cancers, respectively. High expression of insulin/IGF1R positively correlated with a high level of infiltration of macrophages in the tumor, as well as advanced tumor stages [159]. This signaling axis has been implicated in promoting cancer progression, angiogenesis, and metastasis [160]. Moreover, elevated insulin/IGF1R signaling confers drug resistance of breast cancer cells to antiestrogens [161]. In a study of 222 British patients with ER+ mBC treated with tamoxifen, a polymorphism of the IGF1R gene was found to significantly increase the risk of tumor progression and death [162].

Insulin/IGF1R signaling has significant implications for treatment of and survival following breast cancer. The IGF signaling-blocking antibody xentuzumab was tested in a syngeneic model of orthotopically transplanted cancer cells in immunocompetent mice. The findings from this model demonstrated that combining xentuzumab with paclitaxel significantly reduced cancer cell proliferation and metastasis to lung compared with monotherapy [159]. Of note, a phase Ib/II study of xentuzumab found that addition of xentuzumab to everolimus (an mTOR inhibitor)/exemestane (a steroidal AI) did not improve the PFS in patients with nonsteroidal AI-resistant, HR+, HER2− mBC [163]. In an MCF-7 mouse xenograft model, it was also observed that cotreatment with the IGF1R antibody, ganitumab, and PI3K signaling inhibitor, BYL719, was shown to induce tumor regression [164]. Therefore, both xenograft and syngeneic models of IGF1R attenuation in cancer have shown promise for targeting IGF1R in cancer. Unfortunately, clinical trials that assessed the efficacy of IGF1R inhibitors in metastatic HR+ and TNBC demonstrated limited success. A phase II placebo-controlled study found that adding ganitumab to endocrine treatment (exemestane or fulvestrant) in postmenopausal women with HR+ metastatic disease did not improve the PFS but negatively impacted the OS, and hyperglycemia was noted as a common adverse effect of the IGF1R inhibitor [165]. This was not a surprise given that transgenic models of IGF1R attenuation obtained conflicting evidence of increased metastatic foci from mammary gland tumors [166,167] and higher pancreatic tumor grade [168]. Furthermore, there are phase III trials for IGF1R antagonists/inhibitors, including ganitumab, in other cancers (e.g., non-small-cell lung cancer [169], pancreatic cancer [170], and Ewing sarcoma [171]). All these trials have been unsuccessful, indicating that there are tumor-suppressive/antimetastatic functions of IGF1R. There is extensive crosstalk between ER and IGF1R signaling [172,173], and more studies are needed to investigate the benefit of targeting the insulin/IGF1R signaling pathway for mBC treatment [174].

### 3.10. Breast Cancer Stem-Cell-Targeted Therapies

Stem cells have two basic abilities: self-renewal and differentiation into different types of cells. Cancer stem cells (CSCs) are considered to be the major source of drug resistance and cancer recurrence because of their self-renewal ability and differentiating capability into heterogeneous lineages of cancer cells. The link between CSCs and drug resistance was derived from the following observations: CSCs are more resistant to therapy, cancers that possess a similar gene expression profile to CSCs have a worse prognosis, and tumor cells with combined features of stemness, drug resistance, and dormancy have been identified in several cancers [175].

CSCs use multiple mechanisms to survive cancer therapy, remain dormant, and re- awaken themselves after a long time, eventually acquiring resistance to drugs [11,176,177]. CSCs have become a major focus of current cancer research due to their importance in tumor growth, relapse, and metastasis [178]. Breast cancer stem cells (BCSCs) are pivotal promoters for cell proliferation and self-renewal in breast cancer [179]. Studies have found that the proportion of BCSCs in tumors is raised after chemo/radiotherapy, thus increasing tumor heterogeneity and leading to cancer recurrence or therapeutic failure [180]. BCSCs have a high level of plasticity and heterogeneity. In addition, BCSCs overexpress ABC transporters that efflux anticancer drugs out of cancer cells [181], as well as transcription factors, such as FOX2, that promote epithelial-to-mesenchymal (EMT) transition [182], contributing to therapeutic resistance [183]. Therefore, eradicating BCSCs is a promising approach to improving survival or even curing cancer for patients.

EMT transition of breast cancer cells is a key program of generating BCSCs [184]. Several signaling pathways, such as the Notch, WNT, TGF-β, Hedgehog (Hh), and Hippo, are often responsible for activating the EMT program in breast cancer. Thus, these signaling pathways are all known to be essential for the self-renewal properties of BCSCs and are emerging as attractive targets to eliminate BCSCs. The Notch signaling pathway is involved in the process of stem cell self-renewal and other developments. Notch signaling is implicated in a variety of tumorigenic processes including EMT, angiogenesis, maintenance of a hypoxic environment, proliferation, CSC self-renewal, and immunomodulation [185]. Recent studies have confirmed that Notch is closely associated with breast cancer development [186]. Four Notch receptors (Notch-1, Notch-2, Notch-3, and Notch-4) and five related ligands [Jagged-1, Jagged-2 (equivalent to Serrate), and Delta-like-1 (DLL-1), DLL-3, and DLL-4] are all identified to conduct paracrine signaling, primarily expressed in arterial and not venal vessels [187]. Jagged-1 and DLL-4 are crucial ligands for tumor vessel creation affecting cancer cells and neighboring components [186]. Jagged-1 has been reported to promote BCSCs, cancer cell growth and metastasis, and resistance to therapy. Meanwhile, DLL-4 has been shown to involve in the main procedure of tumor angiogenesis [188]. The BRD4/Jagged1/Notch1 signaling pathway is critical for the dissemination of TNBC. BRD4 targeting may be helpful to downregulate this pathway and block TNBC dissemination [189]. A classical therapeutic approach to inhibiting Jagged1/Notch signaling is to use γ-secretase inhibitors (GSIs). Several phase I/II clinical trials have been conducted to evaluate the efficacy of GSIs for solid tumors including breast cancer, showing promising results [190]. GSIs and anti-DLL4 monoclonal antibodies may also have synergistic benefits in blocking the Notch pathway and eliminating the drug-resistant BCSCs in TNBC. Furthermore, the combination of GSIs and chemotherapy or endocrine therapy may enhance overall anticancer efficacy [191,192].

The expression of WNT2, an important gene for metastatic spread, was found enriched in circulating tumor cells released by primary carcinomas [193]. miR-130a-3p, a Notch-regulated ankyrin repeat protein (NRARP) regulator, can inhibit the expression of several genes in the WNT signaling pathway. Overexpression of miR-130a-3p exerted antitumor effects, as evidenced by decreased cell proliferation, anchorage-independent growth, and breast cancer cell migration. Therefore, the tumor-suppressive function of miR-130a-3p in breast cancer may be mediated by inhibiting NRARP and WNT signaling pathway. miR-130a-3p may serve as a therapeutic target for miRNA therapy in breast cancer [194]. Other miRNAs, such as miR-155 [195], miR-140, and miR-22 [196], have been particularly correlated with the regulation of BCSCs properties, thus becoming potential biomarkers and targets for the development of BCSC-based therapy.

Hh signaling has also been shown to regulate BCSCs and may be a potential therapeutic target in BCSCs. Hh signaling inhibitors have been reported to eliminate BCSCs [197]. It was observed that, in trastuzumab-resistant breast cancer cells, Notch-1 negatively regulated PTEN expression, resulting in hyperactivation of MAPK signaling that promoted cell proliferation and survival of BCSCs [198]. Loss of PTEN has been observed in a majority of human breast cancers [199,200]. Therefore, targeting PI3K/AKT signaling may affect the survival of BCSCs. Indeed, the AKT inhibitor, perifosine, has been reported to inhibit the formation of mammospheres in vitro and eliminate breast tumor in vivo [201]. However, a phase II clinical trial that evaluated the efficacy of AKT inhibitor MK-2206 in mBC patients with dysregulation of PIK3CA, AKT1, or PTEN did not reveal any clinical benefit of the AKT inhibitor monotherapy [202].

Many other strategies have also been developed to tackle BCSCs. They include molecular-targeted therapy, differentiation therapy, tumor microenvironment-targeted therapy, and immunotherapy [180]. The stem-cell markers CD44^+^/CD24^−^ have been reported in various breast cancer subtypes and histological stages [203]. A study found that a cell population isolated from malignant human breast cancer-derived pleural effusions was detected with high expression of CD44 and low or no expression of CD24 (CD44^+^CD24^−/low^) [204]. The addition of TGFβ1 drastically increased the CD24^−^ cell populations from the CD24^+^ primary human mammary epithelial cells [184]. Another stem-cell marker, ALDH1, is highly expressed in HER2+ and basal-like tumors. ALDH1 plays a critical role in reducing oxidative stress and resistance to chemotherapies, as well as removing free radicals from radiotherapy [205,206]. The inhibition of ALDH activity with all-trans retinoic acid (ATRA) and specific ALDH inhibitor diethylamino benzaldehyde (DEAB) was successfully reported, sensitizing TNBC cells to chemotherapy and radiotherapy [207]. TGFβ was enriched alongside ALDH^high^ and CD44^+^/CD24^−^ in chemotherapy-treated TNBC patients [208]. TGF-β inhibition could be a promising approach to targeting mesenchymal (CD44^+^/CD24^−^) and epithelial (ALDH^high^) CSCs in TNBC. This has been explored in clinical trials [209]. Moreover, the active DNA damage response and its repair pathway in BCSCs can reduce cell apoptosis or other types of cell necrosis. Since the ataxia telangiectasia mutated (ATM) gene regulates the DNA damage surveillance/repair system, reducing the radiation resistance of BCSCs by targeting ATM signaling may prevent recurrence after routine first-line therapy [210].

### 3.11. Androgen Receptor

The role of AR in breast carcinomas has attracted much attention recently. AR expression is a common feature of both invasive and noninvasive breast cancer, especially in HR− breast cancer [211]. AR expression has been observed in approximately one in three ER−, high-grade, invasive ductal carcinomas [212]. A study that reviewed 980 consecutive cases of invasive breast carcinomas found that AR was notably expressed in ER− breast cancer and TNBC [213]. Many studies have found that activated AR inhibits cancer cell proliferation in most ER+ breast cancers and promotes cancer cell growth in most ER− breast cancers. One study demonstrated that androgen dehydroepiandrosterone sulfate (DHEAS) inhibited cell growth in AR-expressed HR-negative breast cancer cell lines [214], while another showed that iron-regulated transporter-like protein 9 (ZIP9) mediated the androgen-induced apoptosis of TNBC cells [215]. Moreover, treatment with AR inhibitor enzalutamide for HER2+ ER− breast cancer decreases HER2 phosphorylation, and treatment with enzalutamide plus trastuzumab further improves the inhibition of cell growth compared to single drug treatment [216]. Treatment with the non-aromatizable androgen 5α-dihydrotestosterone (DHT) was also reported to significantly inhibit cell proliferation and metastasis in ER+ breast cancer [217]. Some studies have also indicated that AR is correlated with endocrine therapy resistance in breast cancer. Several samples of tamoxifen-resistant breast cancers had a low level of ER expression and a high level of AR expression. Treatment with the AR inhibitor bicalutamide reversed the resistance, indicating that AR signaling is directly involved [218]. However, a phase II single-arm study showed that bicalutamide in combination with a different aromatase inhibitor (AI) did not have a synergistic activity in patients with AI-resistant ER+ breast cancer [219]. Nevertheless, AR has been suggested to be a potential therapeutic target for breast cancer, especially for high-risk breast carcinomas that are AR+/ER− [220]. Bicalutamide and enzalutamide are the most used AR antagonists for tamoxifen-resistant mBC or TNBC [218,221,222]. Clinical trials for these two drugs have achieved excellent results [223,224]. A phase II study analyzing the efficacy and safety of enzalutamide in 118 AR+ TNBC patients exhibited the clinical antitumor function and well tolerance of enzalutamide, thus supporting the addition of enzalutamide for TNBC treatment [224].

### 3.12. Matrix Metalloproteinases and Angiogenesis Inhibitors

Matrix metalloproteinases (MMPs) function as endopeptidases to degrade proteins in the extracellular matrix to facilitate the angiogenesis process [225]. They are involved in breast carcinogenesis, invasion, and metastasis [226]. Overexpression of MMPs is a prognostic biomarker for breast cancer patients, indicating an increased risk of poor prognosis [227] and a higher rate of distant metastases [228]. MMP inhibitors as cancer therapeutics have been investigated in many studies including one phase III trial, which tested the broad-spectrum MMP inhibitor marimastat in mBC, but no therapeutic benefit was found when it was used after first-line chemotherapy [229]. Although results from clinical trials of small-molecule, broad-spectrum MMP inhibitors as cancer therapeutics were disappointing [230,231], MMP2-selective inhibitors were shown to prevent bone metastasis of breast cancer [232]. A recent mega-analysis also suggested that many MMP family members were differentially expressed in patients, and MMP1 and MMP9 were particularly indicated to be potential therapeutic targets [233]. Therefore, more studies are needed to understand how each MMP functions in the tumor microenvironment in order to participate in cancer development and progression.

Antiangiogenic therapy inhibits the formation of new blood vessels needed for the growth of tumors. Bevacizumab was the first anticancer agent to target angiogenesis for cancer treatment. It is a humanized monoclonal antibody against VEGF-A, which is one of the key factors for inducing tumor angiogenesis [234]. While bevacizumab is effective in the treatment for many other cancers, its approval in the mBC indication was revoked on 18 November 2011 by the FDA due to reassessment of its risk/benefit balance following its initial approval on 22 February 2008 [235]. However, bevacizumab remains approved for mBC in the European Union and many other countries. Early clinical trials of bevacizumab in breast cancer patients demonstrated an increase in PSF, but not in OS while exhibiting higher than anticipated toxicity [236,237]. Following the RIBBON-1 phase III trial, the indication was later extended to include first-line treatment of mBC in combination with capecitabine in certain patients [238]. Recent trials, which combined bevacizumab with endocrine therapy and chemotherapy, recommended bevacizumab in selected patients with mBC. In the BOOSTER phase II trial, ER+/HER2− mBC patients who received switch maintenance endocrine therapy plus bevacizumab after fixed cycles of first-line induction chemotherapy had a much longer TFS (time to failure of strategy) than those who received weekly paclitaxel plus bevacizumab [239]. However, in the CALGB 40603 (Alliance) trial, adding bevacizumab to weekly paclitaxel administration followed by doxorubicin and cyclophosphamide did not improve the long-term outcome of TNBC patients, despite a significant increase in the pathologic complete response rate [240]. Therefore, patient selection and bevacizumab treatment schedule may affect the outcome. The efficacy of bevacizumab in breast cancer treatment needs further improvement.

### 3.13. Immunotherapy of Breast Cancer

Cancer immunotherapy utilizes the body’s immune system to recognize and kill cancer cells. Cancer cells use a survival process called immunoediting to evade immune destruction, which is one of the hallmarks of cancer. Immunoediting consists of three phases: elimination, equilibrium, and evasion [241]. During the elimination phase, cancer-specific antigens are recognized and killed via immunosurveillance through the body’s innate and adaptive immune systems. Some cancer cells survive elimination and enter the equilibrium phase, during which surviving cancers are unable to progress but continue to coexist with the immune system for a long time until they develop resistance to the antitumor immune response and escape [242]. Cancer cells develop multiple mechanisms during the evolution of the evasion phase, including alteration or loss of antigens, manipulation of cytokine expression, and upregulation of immune checkpoint proteins [243,244]. These mechanisms are the theoretical bases of cancer immunotherapy. Early approaches to cancer immunotherapy targeted cytokines to affect immune cell function.

Cancer immunotherapies include the use of toxins, tumor necrosis factors, oncolytic vaccines, cytokines, antibodies, immune checkpoint inhibitors (ICIs), oncolytic viruses, and chimeric antigen receptor (CAR) T cells [241]. Currently, ICIs and CAR T-cell therapies are the pioneers of cancer immunotherapy. ICIs have been widely tested in all malignancies, but CAR T-cell therapies have proven to be more successful in hematological tumors than in solid tumors because CAR T cells are given back to the patient via the bloodstream and lymphatic system and have more contact with tumor cells in the bloodstream than in solid tumors. Although ICIs can be easily applied to all malignancies, they are still much more successful in hematological malignancies than in solid tumors due to several barriers, including an immunosuppressive tumor microenvironment, inefficient trafficking, heterogeneity of tumor antigens, and the high intratumoral pressure of solid tumors [245]. Despite these barriers, much progress has been made in delivering immunotherapy to solid tumors [246].

ICIs include monoclonal antibodies that target T-cell-bound negative regulatory proteins: programmed cell death-1 (PD-1), such as nivolumab, pembrolizumab, and cemiplimab; programmed cell death ligand-1 (PD-L1), such as atezolizumab, avelumab, and duravulumab; cytotoxic T-lymphocyte-associated antigen 4 (CTLA-4), such as ipilimumab and tremelimumab [247]. In breast cancer, early studies of ICI monotherapy only demonstrated occasional responses [248] and failed to obtain a survival advantage over chemotherapy [249]. However, it was noted that the response was higher in patients with TNBC compared to other subtypes, especially among those with PD-L1-positive TNBC.

Recent research has investigated the combination of chemotherapy and immunotherapy, i.e., chemoimmunotherapy, for advanced or metastatic TNBC. The interim analysis of the phase III Impassion130 trial [250] showed that, in the intention-to-treat group, patients with untreated advanced TNBC who received nab-paclitaxel plus atezolizumab had a median PFS of 7.2 months compared to 5.5 months for those who received only nab-paclitaxel; among patients with PD-L1-positive tumors, the median PFS was 7.5 months and 5.0 months, respectively. The median OS in the intention-to-treat group was 21.3 months with atezolizumab plus nab-paclitaxel vs. 17.6 months with nab-paclitaxel only, but this was not significant (*p* = 0.08); among patients with PD-L1–positive tumors, the median OS was 25.0 months vs. 15.5 months, respectively. The trial, thus, demonstrated that atezolizumab plus nab-paclitaxel prolonged the PFS by 2.5 months over nab-paclitaxel alone in patients with PD-L1–positive tumors. On the basis of the results of this interim analysis, on 8 March 2019, the FDA granted accelerated approval of atezolizumab and nab-paclitaxel as the first cancer immunotherapy for breast cancer. However, this approval was voluntarily withdrawn on 27 August 2021, after a confirmatory IMpassion131 trial failed to confirm the interim results of the IMpassion130 trial. Both trials were similarly designed except that the IMpassion131 trial used paclitaxel instead of nab-paclitaxel. Unlike the IMpassion130 trial, the IMpassion131 did not meet its primary endpoint of median PFS; the addition of atezolizumab to paclitaxel for treatment of metastatic TNBC did not improve the PFS or OS in either the PD-L1-positive group or the intention-to-treat group [251]. It came as no surprise given that the IMpassion130’s second interim [252] and final [253] OS analyses both still did not find any OS benefit in the intention-to-treat population, despite a clinically meaningful OS benefit with atezolizumab plus nab-paclitaxel in PD-L1 immune cell-positive patients, consistent with that shown in the prior interim analysis [253]. It is still not clear why there was no benefit from the addition of atezolizumab to paclitaxel. However, a follow-up analysis of the IMpassion130 trial suggested that a richer tumor immune microenvironment, as determined by intratumoral CD8 and stromal tumor-infiltrating lymphocyte positivity, was associated with PD-L1 immune cell-positive status and made the combination of atezolizumab with paclitaxel more efficacious [254]. Results from these two trials warrant investigations into the role of atezolizumab for metastatic TNBC.

Another randomized, placebo-controlled, double-blind, phase III clinical trial (KEYNOTE-355) tested the efficacy of pembrolizumab for TNBC. This trial demonstrated that addition of pembrolizumab enhances the antitumor activity of chemotherapy in patients with metastatic TNBC. Additional pembrolizumab treatment extended the median PFS from 5.6 to 9.7 months among patients with a PD-L1 positive score of 10 or more, and from 5.6 to 7.6 months among patients with PD-L1 positive score of 1 or more [255]. This trial suggested that the addition of pembrolizumab to standard chemotherapy for first-line treatment of metastatic TNBC significantly improved survival of TNBC patients. On the basis of the results of this trial, on 13 November 2020, pembrolizumab was granted FDA accelerated approval for unresectable or metastatic TNBC with a PD-L1-positive score of more than 10.

So far, current trials have shown limited success of immunotherapy as a monotherapy in treating mBC, but it remains promising when used in combination with chemotherapy (chemoimmunotherapy). There are ongoing trials evaluating the efficacy of ICIs in combination with targeted therapies, such as PARP inhibitor niraparib, CDK4/6 inhibitor abemaciclib, AKT inhibitor ipatasertib, and MEK inhibitor cobimetinib [248,256]. To improve the outcomes of metastatic TNBC with immunotherapy, in addition to exploring different combinations with other conventional therapies, it is imperative to identify predictive biomarkers for the selection of patients who are more likely to benefit from immunotherapy.

### 3.14. Antisense Oligonucleotide Strategy as a Future Therapy for Breast Cancer Treatment

Antisense oligonucleotide (AON) therapies use short strands of modified nucleotides to target RNA in a sequence-specific manner, inducing the targeted protein knockdown or restoration [257]. The binding of double-stranded or single-stranded AONs to the RNA transcript triggers RISC-mediated or RNaseH-mediated mRNA degradation, respectively [258,259], leading to diminished expression of the target protein. For protein restoration, single-stranded AONs are used to modulate pre-mRNA splicing and either include or skip an exon to restore protein production [260]. AONs can also target noncoding RNAs, such as microRNAs or long noncoding RNAs, to regulate protein-coding gene expression. This therapy has great potential in breast cancer treatment by specifically targeting genes that are overexpressed or mutated in breast cancer cells [261]. It has been verified that the combination of AON targeting miR-21/miR-155 and photodynamic therapy for TNBC could suppress the migration of cancer cells, effectively inhibiting the proliferation and metastasis [262]. A study of the treatment combining antisense BCL-2 with HER2 oligonucleotide in different breast cancer cell lines also showed emerging results, not only suppressing the expression of BCL-2 and pAKT, but also enhancing anticancer drug sensitivity [263]. Similarly, it has been demonstrated that AON targeting Cockayne syndrome group A (CSA) drastically impairs the tumorigenicity of breast cancer cells by hampering their survival and proliferative capabilities without damaging normal cells, which might be a very attractive target for the development of more effective anticancer therapies in breast cancer [264]. However, further clinical evidence of this treatment strategy is still lacking.

## 4. Concluding Remarks

Metastatic breast cancer is a leading cause of morbidity and mortality in women. High intertumoral and intratumoral heterogeneity of cancer means that there is no “one-size-fits-all” treatment for all breast cancers, and that no breast cancer can be treated by a monotherapy. This represents one of the major problems limiting the efficacy of current cancer therapies. Therefore, screening for new therapeutic targets is essential for the development of novel therapies. In the past decades, many novel potential therapeutic targets for treating mBC have been identified through high-throughput approaches. Many of them have been studied in both preclinical and clinical settings. A large body of evidence supports the use of these therapeutic targets for mBC treatment in various clinical trials and offers excellent opportunities for the development of new strategies, especially in combination with standard-of-care treatments. However, TNBC does not yet have an obvious tumor-specific receptor or pathway to target. Therefore, identifying new intrinsic therapeutic targets for TNBC is of high clinical priority in the future. Furthermore, drug resistance is another challenge in cancer treatment. Combining therapeutics to simultaneously target multiple oncogenic signaling pathways may be a key to overcoming/preventing resistance in mBC management and treatment [265].

Tumors are traditionally treated by a combination of conventional therapies including surgery, local radiotherapy, and chemotherapy. The advent of targeted therapy and immunotherapy has resulted in a change in practice of how cancer is treated. As concluded by a recent study that analyzed the link between targeted therapies and outcomes among various types of cancer treatments, only targeted therapies have improved the survival of mBC patients [266]. While targeted and immunotherapy therapies are promising in treating a subtype of disease, they still suffer from the same difficulties as conventional chemotherapy, i.e., lack of effective predictive biomarkers and drug resistance. Biomarkers are needed for patient selection to maximize the efficacy. Overcoming drug resistance is essential to completely cure cancer and prevent it from coming back from residual disease arising from dormant cancer cells hidden in the body. As discussed above, it is evident that a correct combination of different therapies and a right schedule of treatment may improve the efficacy of targeted therapies. Therefore, current standard of care for mBC is chemo-targeted therapy, while chemoimmunotherapy can be catered for certain subtypes of metastatic TNBC.

**Table 1 biology-12-00697-t001:** Therapeutic targets and their targeted agents for metastatic breast cancer.

Targets	Targeted Agents			Patient Population	FDA Approval Year	Reference
	Generic Name	Brand Name	Class			
TROP-2	Sacituzumab govitecan (-hziy)	Trodelvy	TROP-2 ADC	TNBC mBC	2021	[93]
				HR+, HER2− mBC	2023	[96]
CDK4 and CDK6	Palbociclib	Ibrance	CDK4/CDK6 inhibitor	HR+, HER2− mBC	2017	[32]
	Ribociclib	Kisqali	CDK4/CDK6 inhibitor	HR+, HER2− mBC	2017	[32]
	Abemaciclib	Verzenio	CDK4/CDK6 inhibitor	HR+, HER2− mBC	2017	[32]
PI3K/AKT/mTOR pathway	Alpelisib	Piqray	PI3K inhibitor	*PIK3CA*-mutated, HR+, HER2− mBC	2019	[53]
	Buparlisib		PI3K inhibitor	*PIK3CA*-mutated, HR+, HER2− mBC		[55,56]
	Pictilisib		PI3K inhibitor	*PIK3CA*-mutated, ER+, HER2− mBC		[57]
	Taselisib		PI3K inhibitor	*PIK3CA*-mutated, ER+, HER2− mBC		[58]
	Everolimus	Afinitor	mTOR inhibitor	HR+, HER2− mBC	2012	[65]
	Dactolisib		PI3K-mTOR inhibitor	N/A		[68]
	Perifosine		AKT inhibitor	mBC		[202]
PARP	Olaparib	Lynparza	PARP inhibitor	Germline *BRCA* mutations mBC	2022	[72]
	Talazoparib	Talzenna	PARP inhibitor	Germline *BRCA* mutations mBC	2018	[74]
	Veliparib		PARP inhibitor	Metastatic TNBC		[78]
Src pathway	Saracatinib		Src inhibitor	HR− mBC		[115]
	Dasatinib	Sprycel	Src inhibitor	mBC		[118]
	Bosutinib	Bosulif	Src inhibitor	mBC		[120]
Histone deacetylase	Entinostat (MS-275)		Histone deacetylase inhibitor	N/A		[143,144]
	Vorinostat (SAHA)		Histone deacetylase inhibitor	N/A		[145,146]
	Panobinostat (LBH-589)	Farydak	histone deacetylase inhibitor	N/A		[147,148]
HER2	Trastuzumab	Herceptin	HER2 antibody	HER2+ mBC	1998	[18]
	Pertuzumab	Perjeta	HER2 antibody	HER2+ mBC	2017	[21]
	Neratinib	Nerlynx	HER2 small-molecule inhibitor	HER2+ mBC	2020	[24]
	Lapatinib		HER2 small-molecule inhibitor	HER2+ mBC	2007	[154]
	Pyrotinib		HER2 small-molecule inhibitor	HER2+ mBC		[156,158]
BCSCs	γ-Secretase inhibitors (GSIs)		γ-secretase small-molecule inhibitor	N/A		[190]
	Diethylamino benzaldehyde (DEAB)		ALDH inhibitor	N/A		[207]
AR	Enzalutamide	Xtandi	AR inhibitor	AR+ TNBC		[224]
	Bicalutamide	Casodex	AR inhibitor	AR+, ER+		[219]
Insulin/IGF1R pathway	Xentuzumab		IGF antibody	HR+, HER2− mBC		[163]
	Ganitumab		IGF1R antibody	HR+ mBC		[165]
MMP	Marimastat		MMP pan-inhibitor	mBC		[229]
Angiogenesis	Bevacizumab	Avastin	VEGF-A antibody	mBC	2008 (revoked 2011)	[267]
	Ramucirumab	Cyramza	VEGFR-2 antibody	mBC		[268]
ICIs	Atezolizumab	Tecentriq	PD-L1 antibody	mTNBC	2019 (withdrawn 2021)	[250]
	Pembrolizumab	Keytruda	PD-1 antibody	mTNBC	2020	[255]

## Figures and Tables

**Figure 1 biology-12-00697-f001:**
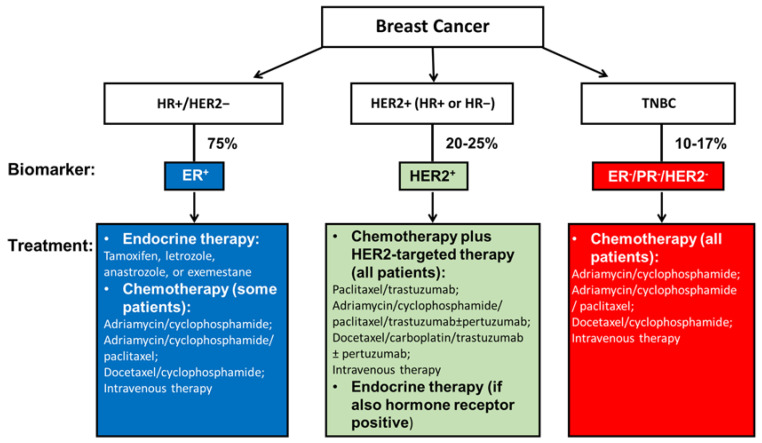
Current systemic management of breast cancer according to tumor subtypes. Current subtype-directed systemic breast cancer treatment consists of endocrine treatment for all HR+ tumors, HER2-targeted therapy plus chemotherapy for all HER2+ tumors, and chemotherapy for TNBC. However, once drug resistance is acquired among endocrine therapy or HER2-targeted therapy for HR+ and/or HER2+ breast cancer patients, the only approach becomes chemotherapy. The same is true for TNBC patients who do not yet have an effective therapy that targets specific receptors. Therefore, innovative and more effective treatment strategies that interfere with specific molecules in TNBC are being investigated, as referenced throughout this review.

**Figure 2 biology-12-00697-f002:**
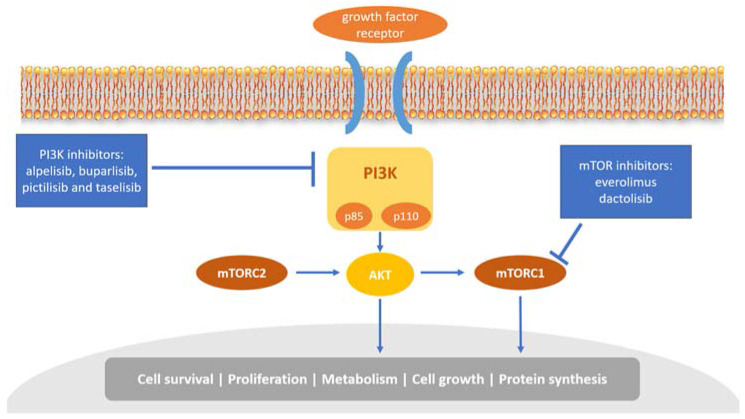
Overview of the PI3K/AKT/mTOR pathway and inhibitors targeting this pathway. PI3K mediates the activation of several protein kinases, including AKT, and then promotes cell survival and cell-cycle progression. mTORC1, the downstream of AKT, regulates proliferation, metabolism, cell growth, and protein synthesis.

## Data Availability

Not applicable.

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
