# Peer review of "Emerging Intrinsic Therapeutic Targets for Metastatic Breast Cancer"

_biology, 2023, doi:10.3390/biology12050697_

Round 1

Reviewer 1 Report (Previous Reviewer 3)

The authors have satisfactorily addressed all queries. I recommend the manuscript for publication.

Author Response

Thank you so much for your approval.

Reviewer 2 Report (New Reviewer)

Dear Authors, your article is well written. I made same remarks which you can find in the attached file.

I have a suggestion for you:

If it is possible to add a paragraph regarding the use of antisense oligonucleotide strategy as a future therapy in the treatment of breast cancer ?

For example:

Dual antisense oligonucleotide targeting miR-21/miR-155 synergize photodynamic therapy to treat triple-negative breast cancer and inhibit metastasis

Antisense oligonucleotides targeted to the human α folate receptor inhibit breast cancer cell growth and sensitize the cells to doxorubicin treatment

Antisense Bcl-2 and HER-2 oligonucleotide treatment of breast cancer cells enhances their sensitivity to anticancer drugs

Antisense Anti-MDM2 Oligonucleotides As a Novel Therapeutic Approach to Human Breast CancerIn Vitro and in Vivo Activities and Mechanisms

CSA antisense targeting enhances anticancer drug sensitivity in breast cancer cells, including the triple-negative subtype

Author Response

Dear Reviewer,

Thank you so much for your kind comments. We have modified the manuscript according to your comments. 

Warm regards,

Jiawei Li

Reviewer 3 Report (New Reviewer)

The review is well written and very extensive. However, they need to include figures for the subsections with various pathways for the readers to understand it better.

it is also better to include the data from the clinical trials (survival and expression of the key targets).

missing references for the statements that delayed child bearing as a contributing factor for increased incidence.

some of the references seems to be review articles instead of the original article. either state it as reviewed in ... or cite the original article.

reference 143 is a xenograft TNBC in mice, its not a mouse model for TNBC.

Author Response

Dear Reviewer,

Thank you so much for your kind comments. We have modified the manuscript according to your comments. 

Warm regards,

Jiawei Li

Round 2

Reviewer 3 Report (New Reviewer)

none

This manuscript is a resubmission of an earlier submission. The following is a list of the peer review reports and author responses from that submission.

Round 1

Reviewer 1 Report

Please provide a detailed outline of what this review adds that is not already present in reviews of metastatic breast cancer

Reviewer 2 Report

Overall Summary

In their manuscript “Emerging Intrinsic Therapeutic Targets for Metastatic Breast Cancer,” Li and colleagues aim to provide an inclusive look into emerging breast cancer therapies. These include drugs that are recently FDA approved, such as PARP inhibitors, as well as drugs that are struggling in clinical trials, such as IGF1R inhibitors. The authors introduce each drug type with important information about how they work and what their approval status is. Overall this is an interesting and informative review. Major strengths are coverage of new, major advancements in breast cancer therapy and a focus on triple-negative breast cancer therapy, since this subtype is most in need of new treatments. The graphics are simple yet informative. There are some weaknesses that should be addressed to increase the impact of the review. While some sections are well-written, others need major grammar corrections, some of which are listed below. Often the grammatical mistakes obscure the intended meaning. The manuscript should be reviewed in its entirety for grammar and English usage. In addition to corrections to the writing, more context could be provided in some sections. For instance, in the “Targeting Insulin/IGF1R” section, it should be mentioned that ganitumab, among other drugs, has entered phase 3 trials for other cancers and failed, initiating an investigation into IGF1R and its tumor suppressive functions.

Specific Points to Address

1)    Lines 18-19: I suggest changing “Novel therapies that targeted these targets” to something such as “Novel therapies against these targets” to reduce redundancy.

2)    Simple summary needs some grammatical revision, such as removing “nature” from line 21.

3)    In the graphical abstract, xentuzumab and ganitumab are both labeled as IGF1R antibodies, however xentuzumab is an IGF ligand inhibitor that is described as an IGF1R/insulin signaling blocker in the text body. I would relabel as “IGF1R/InsR inhibitors” (InsR=insulin receptor) in the graphical abstract to cover these bases.

4)    Line 41: Mechanic -> mechanistic

5)    Line 78: “Monoclonal antibody, also known as therapeutic antibody” is a sentence fragment, I recommend a grammar check for this paragraph.

6)    Line 91: I would briefly state what the three main tumor subtypes are here. You list them out in Figure 1, which is good, but it is not clear in the text.

7)    Line 97: This is a good contextualization of the problem. I would also add in a sentence or two about cancer cells remaining dormant/latent in bone marrow, as this is a major cause of cancer recurrence: https://www.nature.com/articles/s12276-020-0423-z. This might be a good addition to the cancer stem cell section as well.

8)    Line 101: “assumes to achieve” is awkward wording, perhaps say “aims to achieve.” I suggest grammar-proofing this paragraph as well.

9)    Lines 134-136: Should reiterate that chemotherapy alone reduces quality of life of patients, and combination therapy with targeted therapeutics aims to increase quality of life by reducing reliance on cytotoxic chemotherapy alone.

10) Line 135: State here that progression free survival will be abbreviated as PFS going forward.

11) Line 151: I suggest briefly stating the mechanism of action of trastuzumab, since the mechanism of pertuzumab is stated but it is unclear how the two drugs differ.

12) Line 174: Subsequently -> consequently. I suggest checking the grammar of the various transitional words used, such as besides.

13) Lines 243-244: This sentence needs to be reworked. PIK3CA mutations are gain-of-function in cancer since the catalytic domain of PI3K is oncogenic: https://breast-cancer-research.biomedcentral.com/articles/10.1186/bcr3605#:~:text=PIK3CA%20mutations%20represent%20one%20of,factor%20receptor%202%2Dpositive%20subtypes.  “Loss of PIK3CA mutations” should read “Acquisition of PI3KCA mutations.” Also, “exits” -> exist in line 243.

14) Lines 256-271: Grammar/spell check this paragraph, eg. primiray -> primary in line 268.

15) Lines 272-274: First sentence is a bit awkward, consider rewriting, eg: “since its discovery, the mTOR molecular pathway has been implicated in the pathogenesis of breast cancer and has become a prominent therapeutic target.” The rest of the manuscript requires a close grammar and spell check.

16) Line 293: Potential -> promising

17) Line 315: The hyphens in this line are confusing since one is a hyphenation and the other means “negative,” so I suggest changing “BRCA-mutation” -> “BRCA-mutant” and writing out “Her2-negative” to clarify that the patients are BRCA mutation positive, Her2-negative.

18) Line 342: Unsure what is meant by “further.”

19) Line 371: I suggest briefly stating what was different about bosutinib’s safety profile versus dasatinib. Was it better or worse? For instance, the article states that dasatinib elicited higher rates of hematological complications than bosutinib.

20) Line 379: I would use a word other than “differentiation” here, since I do not think you are discussing cell differentiation in this context. If you are mentioning cell differentiation, another source is needed as the citation does not mention the effect of JAK/STAT signaling on cell differentiation.

21) Line 422: Need to write out the APCL acronym before abbreviating and give a brief explanation of what it is.

22) Section 3.7: Although the activated growth factor receptor and IGF1R/insulin sections are listed separately, IGF1R and InsR are growth factor receptors. I suggest changing the heading of section 3.7 to “Inhibition of EGFR/Her2” and make the IGF1R/insulin section 3.8 to make both sections adjacent and improve organization.

23) Line 489: “which increases the propensity for metastasis gene” perhaps should read “which increases pro-metastatic gene expression.”

24) Line 507: Since PTEN is the repressor of PI3K/AKT signaling, rewrite “PTEN/AKT signaling inhibitors” -> “PI3K/AKT signaling inhibitors.”

25) Line 511: Add “results,” ie “obtained poor results”.

26) Lines 562-563: “correlates with increased macrophage infiltration and advanced tumour stage,” is taken word for word from the citation text, please change wording.

27) Lines 564-565: While correct, it is better to state how IGF1R signaling affects antiestrogen sensitivity, ie positively or negatively correlated. The citation states that elevated IGF1R/insulin signaling confers resistance to antiestrogens.

28) Lines 570-572: Content here is fine but again, the statement “reduced incidence of metastasis, and a significant reduction of tumour cell proliferation and metastatic burden compared with monotherapy” is nearly identical to source material, please reword.

29) Lines 570-573: I suggest noting that in this study, these mouse models were syngeneic models of orthotopically transplanted cancer cells in immunocompetent mice. Xenograft and syngeneic models of IGF1R attenuation in cancer have shown promise for targeting IGF1R in cancer; however, transgenic models of IGF1R attenuation exist that show conflicting evidence of increased metastatic foci from mammary gland tumors (https://www.ncbi.nlm.nih.gov/pmc/articles/PMC4782979/ and https://breast-cancer-research.biomedcentral.com/articles/10.1186/s13058-018-1063-2 ) and higher pancreatic tumor grade (https://pubmed.ncbi.nlm.nih.gov/18451178/ ).

30) Lines 573-574: I suggest briefly describing the model used: MCF7 mouse xenograft.

31) Line 578: I suggest briefly mentioning that hyperglycemia was a common adverse effect of the IGF1R inhibitors, as reported in reference 152.

32) Line 579: IGF-1R -> IGF1R.

33) There are several instances where a single sentence makes up a paragraph. These should be incorporated into the prior or subsequent paragraphs.

34) Section 3.10: I recommend mentioning that there are phase 3 trials for IGF1R antagonists/inhibitors, including ganitumab, in other cancers that have been unsuccessful, indicating that there are tumor suppressive/antimetastatic functions of IGF1R.

35) Suggested Questions to Address

Three other types of cancer-intrinsic targeted therapy being developed are matrix metalloproteinase, angiogenesis and integrin inhibitors. One or more of these would be interesting to cover, even if briefly. For instance, a clinical trial exists for the use of bevacizumab (angiogenesis inhibitor) in the treatment of breast cancer: https://clinicaltrials.gov/ct2/show/NCT00445406?term=matrix+metalloproteinase&cond=breast+cancer&draw=2&rank=3.

Reviewer 3 Report

In the current manuscript, the authors have summarized the major chemotherapeutic treatments available for all breast cancer subtypes. The authors have also explained the mechanisms of action of inhibitors targeting specific receptors that are over-expressed in metastatic breast cancer. However, there are several shortcomings that the authors need to address to make the manuscript suitable for publication.

1.     In the section on targeting CDK4 and CDK6, the authors have alluded to the fact that retinoblastoma-associated proteins are phosphorylated by the complex of cyclin D and CDK4/CDK6. There are recent publications that deliberate on how the expression of RB-associated proteins affects the incidence of the different subtypes of breast cancer (Patel et al.; (2020) NPJ Breast Cancer, 6, 19). The authors should briefly introduce the correlation between the expression of RB-associated proteins and the development of mBC.  

2.     The authors have described the role of GRAMD1B as an important signaling component of the JAK/STAT pathway and the stimulation of IL6 increases its expression. There are reports about the low expression of interleukin IL-17 inhibiting STAT3 activation (Ma, M. et al. (2018), Int Immunopharmacol, 59, 148-156) and IL-32 theta mediated disruption of the CCL18/STAT3 pathway by preventing the association of CCL18 with its receptor, resulting in the slowdown of the progression of macrophage associated breast cancer (Pham,T.H. et al. (2019), Cell Commun Signal., 14, 7). It is also known that the WW domain containing oxidoreductase can disrupt the interaction between IL-6R and STAT3 that reduces cancer progression (Chang, R.X. et al. (2018), Nat. Commun 9, 12). The authors should include a broader description about the role of regulation of the STAT3 pathway with the different interleukins in slowing the development of mBC.

3.     The role of immunotherapy in the treatment of breast cancer is completely omitted from the manuscript. There are a lot of recent reports on the use of immunotherapy in combination with chemotherapy for the treatment of metastatic breast cancer and TNBC. The authors should provide some insights into the comparative efficacies of immunotherapy, radiotherapy, and chemotherapy in the treatment of mBC.

4.     The following sentences in the manuscript seem to be incomplete and it is not clear what the authors are trying to convey.

a.     2nd paragraph in the introduction, (line 78) – “Monoclonal antibody also known as therapeutic antibodies….”

b.     2nd paragraph in the section on current management of breast cancer (line 103)- “Systemic therapy includes postoperative (adjuvant) and preoperative (neoadjuvant) …..“

c.     3rd paragraph in the section on Breast Cancer Stem Cell targeted treatments (lines 509-511) – “However, a phase II clinical trial for mBC patients with dysregulation of PTEN/Akt/PI3K has evaluated the efficacy of the inhibition of Akt protein obtained poor….”

The authors should reword the above sentences and make them more comprehensible for the reader. Also, a list of all abbreviations used in the manuscript will be helpful for the reader.